# Pain Assessment with the BPS and CCPOT Behavioral Pain Scales in Mechanically Ventilated Patients Requiring Analgesia and Sedation

**DOI:** 10.3390/ijerph191710894

**Published:** 2022-09-01

**Authors:** Katarzyna Wojnar-Gruszka, Aurelia Sega, Lucyna Płaszewska-Żywko, Stanisław Wojtan, Marcelina Potocka, Maria Kózka

**Affiliations:** 1Department of Clinical Nursing, Faculty of Health Sciences, Collegium Medicum, Jagiellonian University, 31-501 Kraków, Poland; 2Department of Anaesthesiology and Intensive Therapy, University Hospital in Kraków, 30-688 Kraków, Poland

**Keywords:** sedation, analgesia, pain, BPS, CCPOT, Intensive Care Unit

## Abstract

Background: Intensive Care Unit (ICU) patients often experience pain, especially during diagnostic, nursing, and therapeutic interventions. Pain assessment using the Behavioral Pain Scale (BPS) and Critical Care Pain Observation Tool (CCPOT) are recommended, but they are difficult to do in patients undergoing deep sedation. This study analyzed the usefulness of the BPS and CCPOT scales in assessing pain among patients with varying degrees of sedation. Methods: In 81 mechanically ventilated and sedated ICU patients, 1005 measurements were performed using the BPS and CCPOT scales. The study was conducted by 3 trained observers 3 times a day (each measurement at rest, during painful nursing interventions, and after the intervention). The Richmond Agitation-Sedation Scale (RASS), the Simplified Acute Physiology Score (SAPS II), and the Acute Physiology and Chronic Health Evaluation (APACHE II) were also analyzed from medical records as well as information on the length of hospitalization and treatment. Results: It was shown that signs of pain increased significantly (*p* < 0.001) during interventions in patients on both scales (BPS and CCPOT), and then returned to values close to the resting period. RASS results correlated significantly (*p* < 0.05) and positively with the results of the BPS and CCPOT. A strong correlation was found between the results of both scales at each stage of the study (R = 0.622–0.907). Conclusions: Nursing procedures are a source of pain in analgosedated patients. The BPS and CCPOT scales are useful tools for assessing the occurrence of pain in mechanically ventilated patients, including those in deep sedation.

## 1. Introduction

In the Intensive Care Unit (ICU), emphasis is placed on the control of pain, agitation, delirium, patient immobility, and sleep (PADIS) [1]. PADIS is an extended concept based on the 2013 guidelines, which state the occurrence of the so-called ICU triads, meaning pain, agitation, and delirium (PAD) [2]. Failure to treat the above conditions worsens the effectiveness of therapy, causing unnecessary suffering for patients and negatively impacting patient quality of life after ICU discharge [1,3,4]. Studies have shown that patients hospitalized in the ICU often experience pain, while at rest [5,6,7] but especially during diagnostic, care, and treatment activities [5,6,7,8,9,10,11,12,13,14]. Unrelieved pain can result in chronic pain, posttraumatic stress disorder symptoms, and lower quality of life. To provide optimal pain management to ICU patients, accurate routine pain assessments is recommended [3]. Better outcomes were found after the implementation of pain assessment tools including a reduction in the duration of mechanical ventilation and ICU stay [15,16].

In line with the 2018 Clinical Practice Guidelines for the Prevention and Management of Pain, Agitation/Sedation, Delirium, Immobility, and Sleep Disruption in Adult Patients (PADIS) in the ICU the most accurate and reliable pain assessment tool among patients unable to communicate with the therapeutic team is still the Behavioral Pain Scale (BPS) and the Critical Care Pain Observation Tool (CCPOT) [1,17]. In addition, new scales are available, including the Facial Action Coding System (FACES), the Pain in Advanced Dementia (PAINAD) and the Behavior Pain Assessment Tool (BPAT), which, however, require further research [1]. It was also upheld that mild sedation as assessed by the Richmond Agitation Sedation Scale (RASS, −2/+1) was more beneficial for the patient, then deep sedation [1]. There is a need for further research on the relationship between the depth of sedation and the perception of pain, sleep, and delirium in patients [18].

In order to standardize the treatment and care of ICU patients, the ABCDEF algorithm is used (spontaneous Awakening, Breathing coordination, attention to Choice of sedation, Delirium monitoring and early mobility and Exercise, Family engagement and empowerment). It involves pauses in sedation, shallow sedation with optimal analgesia associated with spontaneous breathing attempts, early extubation, and mobilization of a cooperating patient, delirium control, and family involvement in care [19,20]. Similar management in ICU patients is recommended according to the concept of e-CASH (early Comfort using Analgesia minimal Sedatives and maximal Humane care), whose main assumptions are using optimal analgesia, eliminating, or maintaining sedation as low as possible, promote sleep hygiene, early mobilization of patients and patient-centered care [21].

The authors of the e-CASH guidelines note, however, that deep sedation is justified in certain groups of patients, such as those with severe respiratory failure with dyssynchrony, during the administration of muscle relaxants, in an epileptic state, in surgically treated patients requiring absolute immobilization, and in those with severe brain damage with intracranial hypertension. Sedation in favor of analgesia should be avoided in the remaining groups of patients [21]. There are few publications in the literature that analyze the occurrence of pain in deeply sedated patients and how to assess it. For this reason, we conducted a study aimed at assessing pain in ICU patients subjected to mechanical ventilation using the BPS and CCPOT behavioral pain scales, including patients under deep sedation (−4/−5 on the RASS scale).

## 2. Material and Methods

The observational study was carried out in 2017–2019 in the Intensive Care Unit of the University Hospital in Krakow after obtaining the consent of the bioethics committee (consent No. 1072.6120.161.2017). The study group consisted of 81 patients-34 (41.97%) women and 47 (58.03%) men aged 56 to 75.8 (mean age 63.1 ± 17.21). The study included adult patients unable to self-report pain, requiring sedation (RASS less than or equal to −1) and analgesia, mechanically ventilated, hemodynamically stable, and staying in the ICU for at least 48 h. The study excluded patients with paresis or paralysis of the upper and/or lower limbs, receiving neuromuscular blocking drugs, after sudden cardiac arrest (SCA), and after injuries that prevented pain assessment (e.g., in the craniofacial region).

The researchers did not intervene in the treatment regimen of the study patients. Prior to our study, monitoring of sedation and analgesia with standardized tools had not been routinely used in the studied ICU. Analgesia and sedation in patients were considered individually and adjusted to the patient’s clinical condition, as stated in the DAS guideline [22], and in surgical patients, analgesic treatment using algorithms developed by Polish pain relief teams were applied [23]. Deep sedation (according to eCASH guidelines) [21] was used in the following patient groups: with severe respiratory failure and dyssynchrony, in surgical treatment requiring absolute immobilization, and in cases of severe brain injury with intracranial hypertension. Analgesics were administered by continuous infusion according to doses selected individually for each patient. For the purposes of the study, no additional boluses of analgesics were administered during any stage of observation.

The study used behavioral pain scales, meaning the BPS, CCPOT and RASS scale (Richmond Agitation-Sedation Scale) to assess the depth of sedation. In addition, data from medical records were analyzed, such as patient records, information on the length of hospitalization, sedation and analgesics, and the risk of death scales through the Simplified Acute Physiology Score (SAPS II) and Acute Physiology and Chronic Health Evaluation (APACHE II).

The Polish version of the CCPOT is a useful and reliable tool for analyzing pain in critically ill, intubated patients using analgosedation without hypnosis protocol based on opioid infusions [24]. The Polish version of the BPS also shows reliable internal consistency (Cronbach’s alpha 0.6883) and is recommended for pain assessment in sedated, mechanically ventilated patients [25]. In our study, pain was defined as BPS score ≥ 5 and CCPOT ≥ 3 [24,26,27].

The study was conducted simultaneously by three independent observers trained theoretically and practically in the uniform use of the BPS and CCPOT scales. The researchers did not consult the results of the assessment with each other. The observation period was 24 h (randomly selected from the entire stay in the ward), during which pain was assessed three times a day (in the morning from 6.30 a.m. to 9.00 a.m., in the afternoon from 1.00 p.m. to 3.00 p.m. and in the evening from 7.00 p.m. to 10 p.m.) in the patients (during rest, nursing interventions, and after an intervention).

Rest meant no medical or nursing intervention for at least 30 min. Interventions in which pain was assessed included the procedure of evacuating secretions from the bronchial tree, changing the dressing on a surgical wound, or repositioning the patient in bed. The evaluation after the intervention was carried out for dozen to several dozen minutes without stimulation from external stimuli. Since in some patients, during the scheduled observations, there were disrupting factors, such as withdrawal of sedation on the day of measurement, diagnostic and surgical procedures with the use of muscle relaxants, in-hospital transport for tests or procedures, the analysis finally included 1005 measurements meeting all the correct criteria.

## 3. Statistical Analysis Methods

The R CRAN software for Windows (version 4.0.2, created by Bengtsson H., Jacobson A. and Riedy J., Vienna, Austria) was used for statistical analysis. The analysis of quantitative variables (raw scores of BPS, CCPOT and RASS) was performed by calculating the mean, standard deviation, median, quartiles, minimum and maximum. The analysis of qualitative variables (scores of BPS and CCPOT cut into two intervals: no pain and occurrence of pain) was performed by calculating the number and percentage of occurrences of each value.

The comparison of the values of the qualitative variables in the groups was performed using the chi-square test (with Yates’ correction for 2 × 2 tables) or Fisher’s exact test, where low expected frequencies appeared in the tables. The comparison of the values of quantitative variables in the two groups was performed using the Mann-Whitney test. Correlations between quantitative variables were analyzed using the Spearman correlation coefficient. Non-parametric methods were chosen since all quantitative variables were not normally distributed, by definition (raw scores of BPS, CCPOT and RASS could take only few distinct values).

The analysis of the influence of quantitative variables on a dichotomous (binary) variable was performed using the logistic regression method. The results are presented in the form of OR (odds ratio) with a 95α% confidence interval. The level of significance was *p* < 0.05.

## 4. Results

The hospitalization length of the studied patients ranged from 11 to 44 days (mean 32.7 ± 35.47 days). On the APACHE II scale, the respondents obtained from 20 to 29 points, on average 25.43 ± 9.39 points in the assessment of the severity of their condition and 30–55 (average 42.73 ± 24.38) points in assessing the risk of death. The SAPS II scores, however, came out to 48–70 (average 59.3 ± 16.57) points and 41–85 (average 60.96 ± 24.42) points respectively. The risk of death increased significantly with the age of the studied patients (OR 1.062; 95% CI: 1.025–1.101; *p* < 0.001). Moreover, the length of hospitalization negatively correlated with the SAPS II scores (*p* < 0.05). The more severe the patients’ health condition and the higher the risk of death on this scale, the shorter the hospitalization, often ending in death. A similar relationship has not been demonstrated with the APACHE II scale. Most of the patients required deep sedation (Figure 1).

Analgesia and sedation in patients were tailored to the patient’s clinical condition. Among the analgesics, Oxycodone and Ketamine were used and Buprenorphine transdermal patch (mostly during the later period of hospitalization). The sedating drugs (apart from Ketamine) used in the study group were Propofol and Midazolam, and occasionally neuroleptics were also administered (Haloperidol-1–2 mg i.m 1–3x/day). Data on average doses of analgesics and sedatives used intravenously in continuous infusion in the study group of patients are presented in Table 1.

In the vast majority of measurements, patients had no signs of pain, except during the interventions. The no-pain scores were very similar on both scales (95.42% and 96.52%, respectively, before and 93.23% and 95.33% after the intervention). In about 1/3 of the measurements, signs of pain were observed in patients during the intervention (Table 2).

Comparing the mean scores before, during and after the nursing procedures from all measurements for both scales (BPS and CCPOT), it was shown that the signs of pain increased significantly during interventions among patients (*p* < 0.001), and then returned to values close to resting during third observation (which took place dozen to several dozen minutes after intervention, Table 3).

The RASS scores correlated significantly (*p* ˂ 0.05) and positively with the BPS and CCPOT scores at all stages of the study (Table 4), which means that patients undergoing deep sedation showed fewer signs of pain.

Moreover, it was found that the degree of sedation of patients, meaning the RASS score, was a significant (*p* < 0.05) independent predictor of an increase in pain intensity during an intervention. The regression parameter for the BPS was 0.254, so the higher the RASS score (shallower sedated patient), the more visible signs of pain during an intervention were observed (on average 0.254 points on the BPS scale per one RASS point). In the case of the CCPOT, the regression parameter is 0.363, so for each RASS point, there was an average increase of 0.363 points in pain. Yet, no differences were found related to the supply of specific analgesics and sedatives (*p* > 0.05).

A strong correlation was demonstrated between the results of both scales at each stage of the study (Table 5).

Analyzing the obtained data in the subgroups of women and men, it was found that men showed not significantly higher scores than women, both before (*p* = 0.055 for the CCPOT and *p* = 0.018 for the BPS) and after interventions (*p* = 0.054 for the CCPOT and *p* = 0.087 for the BPS). On the other hand, the results of the scales during the painful procedures were similar in both sexes (*p* > 0.05).

The age of the respondents correlated negatively with the results of the BPS and CCPOT scales, except during interventions. However, only in the case of the BPS scale, a statistically significant negative correlation between its results before the procedures and the age of the patients was demonstrated (R = −0.073; *p* = 0.027). Thus, the intensity of the signs of pain was not related to the age of the patients during the procedures, but younger patients manifested pain of a slightly greater intensity than older patients before and after the interventions (especially before them, see Table 6). It should be noted that disease diagnoses (meaning the percentages of people after injuries, potentially more prone to pain) were similar in all age groups.

Moreover, the results of both scales during and after the procedures positively correlated with the length of patient hospitalization (Table 7).

The influence of the time of day on the occurrence of pain in the studied patients was also analyzed. Higher results during evening measurements were found in the case of the BPS during an intervention (*p* = 0.043) and in the case of the CCPOT, both before the intervention (*p* = 0.03) and during the intervention (*p* = 0.03, Figure 2).

## 5. Discussion

Pain is one of the most stressful event and common traumatic memories for patients in the intensive care unit (ICU) [8]. Unrelieved pain can have numerous negative consequences for patients [3,28].

The BPS and CCPOT are scales recognized for pain testing in patients unable to verbalize pain, in line with the PADIS [1] recommendations. The BPS and CCPOT scales show very good psychometric properties when used in most ICU patients [29,30], including patients with cerebral trauma [10,31,32,33] and delirium [34]. However, their validity concerning some groups of patients, e.g., with burns or cognitive deficits, requires further research [35].

There are also differences in opinions about the validity of using these scales in unconscious and deeply sedated patients. According to some authors, both of these scales are equally useful for testing pain in patients with varying degrees of analgesia or sedation [36], both conscious or unconscious [37]. This was also confirmed by the studies conducted by Severgnini et al. [29] in which the level of patient awareness did not affect pain detection during nursing procedures, and the combined use of the BPS and CCPOT gave better sensitivity compared to each of these scales separately. This suggests that nursing procedures are a source of pain regardless of the level of sedation [37], and combining the BPS and CCPOT may be a valuable tool for pain assessment in critically ill mechanically ventilated patients [8,29].

According to other authors, however, there are differences in the results of these scales of conscious and unconscious/sedated and unsedated patients, both before and after nursing procedures [7,9]. The heterogeneity of the above data has inspired our research in this area. Another premise for undertaking the research was the fact that the previous studies using the BPS and CCPOT scales were usually conducted on mildly sedated or not sedated patients. For example, Chanques et al. tested the BPS and CCPOT scales among intubated and non-intubated patients who could not verbalize pain, and whose RASS was −3 and above [8]. In turn, Puntillo et al. [9] studied patients with a RASS from −1 to 0. The Polish version of the CCPOT was validated in the ICU where a no-hypnoticanalgesia-based protocol was implemented in the group of intubated patients with a RASS of-3 or greater [24].

Pain scales are usually tested by comparing the results between a painful and a painless procedure or at rest, as in the study by Rijkenberg et al., where the mean difference between painful and painless procedures was 3.13 ± 1.56 (*p* < 0.001) [11]. In our research, we chose turning the patient, aspiration the bronchial tree or change the wound dressing as the painful procedures, which were considered to be ones of the most painful nursing procedures in multicenter studies from 28 countries [9]. Comparing the BPS and CCPOT results during these procedures with the at-rest results and after the procedures, we also came up with a statistically significant difference (*p* < 0.0001). Sedated patients, including some profoundly sedated, with a RASS score of less than −3 showed signs of pain during painful procedures on both the CCPOT and BPS scores, with less sedation being a significant independent predictor of increased signs of pain.

A separate issue is the matter of deep sedation in our group of patients. Sedation and analgesia are undoubtedly important elements in the treatment of mechanically ventilated patients in the ICU. Analgosedation provides patients with comfort, improves patient-ventilator synchrony, reduces anxiety and agitation [38], allows one to perform invasive procedures and reduces stress and oxygen consumption.

On the other hand, studies indicate that the use of excessive sedation is a disadvantage. Deep sedation is associated with a prolonged duration of mechanical ventilation, longer stay in the ICU [18,39], is a risk factor for delirium [40] and causes long-term adverse psychological sequelae, [18,41] and leads to a reduction in patient survival [42]. According to reports, the use of lighter sedation (compared to deep sedation) does not increase the incidence of adverse effects [4,43,44]. The current recommendations of the Society of Critical Care Medicine prescribe the maintenance of light sedation in all patients undergoing mechanical ventilation while recognizing that it is a conditional recommendation due to the low quality of the available evidence [1,45]. Studies by Wang et al. [46] indicate that over 87% of clinicians use analgosedation in ICU patients, and more than half never apply strategies for keeping the patient conscious.

According to the e-CASH guidelines, deep sedation is justified in some patients, despite potential side effects e.g., in patients with ARDS and dyssynchrony, treated with muscle relaxants, in surgical patients requiring immobilisation, and with severe brain injury with intracranial hypertension [21], i.e., in the majority of our study patients.

The two most validated and reliable sedation scales are recommended for use in ICU mechanically ventilated patients-the Sedation Agitation Scale (SAS) and the Richmond Agitation Sedation Scale (RASS) [47]. We used the latter in our research. Despite the reproducibility and clarity of the scales themselves, there is no defined cut-off point for “light” sedation. Some studies define deep sedation as a RASS score from −3 to −5 [3,6] and others as a RASS score of −4 or −5 [14]. It should be emphasized that the cut-off point is an important clinical differentiation, as in the RASS scale a result of −3 means that the patient still responds to a voice, while the results of −4 or −5 show that the patient is unresponsive to a voice and is often in a coma [45]. This also has an impact on the pain assessment.

The use of deep sedation, although it is not recommended, is a rather common ICU phenomenon in many countries [28,38]. As shown by multicenter studies conducted in European countries, organizational factors (the size of the ICU, the number of staff per patient, teamwork) influence the way sedation is used in patients [48]. The preferences of a given center and physicians, cause large differences in the approach to sedation of critically ill patients, hence the need to optimize sedation and analgesia practices with its consequences [49,50].

In our study, most of the patients were deeply sedated (assuming a RASS below −3). ICU practices related to long-term analgosedation (>24 h) were rather typical for Polish hospitals and included opioid, Midazolam and Propofol treatments [17]. Our patients required sedation and long-term mechanical ventilation, mainly due to ARDS, some of which showed dyssynchrony with the work of the ventilator. Also, a fairly large number of trauma surgical patients and patients with intracranial hypertension was deeply sedated in addition to being under analgesia. In patients with these characteristics, pain monitoring is particularly important, so we included them in the analysis to determine whether the BPS and CCPOT will also allow the detection of signs of pain in patients under deep sedation. We excluded patients from the research who were in conditions that could make the expression of pain impossible, regardless of sedation; for example, people with paresis, paralysis of the limbs, receiving neuromuscular blocking drugs, or after craniofacial injuries.

In general, pain was well controlled in the studied patients, because apart from the interventions, they did not show any signs of it according to the adopted criteria (93.2–95.4% on the BPS scale and 95.3–96.5% on the CCPOT scale, respectively). Our study confirms the usefulness of the BPS and CCPOT scales in the assessment of pain in the studied group of patients. Signs of pain, also in patients with RASS scores below −3, were much more visible to independent observers when performing painful procedures in patients than in patients at rest (*p* ˂ 0.001), and returned to values close to resting sometime after the intervention was over. This was confirmed by the simultaneous use of the BPS and CCPOT scales and a strong correlation between the results of both scales at each stage of the study, especially evident during painful interventions (r = 0.907).

Our results show a similar trend to the data obtained by Rijkenberg et al., who found that both BPS and CCPOT scores increased significantly during nursing procedures and returned to baseline levels in a short time [51]. In our research, this increase in scores during painful procedures was smaller (about 1 point), but statistically significant. This could be due to differences in the selection of patients for the study, deeper sedation of the studied patients (and thus a less pronounced manifestation of pain) and a much larger number of measurements. In turn, the correlation between the scales was stronger in our study. A similar correlation between the BPS and CCPOT was demonstrated, in the reports of Liu et al. [52] and a literature review by Birkedal et al. [53].

The severity of pain during painful procedures was more evident in patients who were hospitalized longer in the ICU, which may be explained by the cumulative effect of negative stimuli along with the length of stay in the ward (e.g., fatigue, sleep deprivation, excess stimuli from the ICU environment), which may affect pain experience [54]. Perhaps in a similar way (the amount of stimuli and activity to which patients are exposed during the daytime hours) the clearer signs of pain at the end of the day could be explained compared to the time before noon [55].

Behavioral pain scales do not allow the assessment of pain sensations in patients who cannot manifest it in any visible way, for example, in patients with limb paralysis or craniofacial injuries. It should be assumed that they also experience pain during nursing, diagnostic and treatment activities (although confirming this would require the use of other methods, e.g., pupillometry), so effective methods of minimizing pain should be implemented.

## 6. Study Limitations

The results of our study, based on more than 1000 measurements by three independent observers, seem to confirm the usefulness of the BPS and CCPOT scales in assessing pain in patients undergoing sedation, including deep sedation, yet they have some limitations. The research was conducted in one center (the Trauma Centre for Emergency and Disaster Medicine), which treated the most seriously ill patients-usually those with multiple and multi-organ injuries, requiring highly specialized monitoring, treatment (often surgical interventions) and nursing care in which the analgosedation treatment regimen may have differed from that adopted in other ICUs. Therefore, it was not possible to extrapolate the obtained results. Moreover, the profile of the studied ward, large differentiation among patients in terms of disease entities, the related individual selection of analgesia and the interactions of sedatives and analgesics made it impossible to analyze the relationships between the type and doses of analgesics and the results of the BPS, CCPOT and RASS scales. Further research is required.

## 7. Implications to Practice

As there are no universal signs of pain, and individual pain management based on different tools is a complex task, it should be entrusted to team members after prior training. In pain assessment, tools should be carefully selected according to the patient’s health situation [3,56]. It seems that pain monitoring using the BPS and CCPOT scales among sedated, mechanically ventilated patients is sensitive and reliable, and should be a routine element of ICU management (according to the ABCDEF algorithm and e-CASH guidelines), including regular nursing practice. One should be aware that patients who not only cannot verbalize pain, but also do not manifest it in a manner included in the behavioral pain assessment scales, probably also experience it during nursing, diagnostic, and therapeutic activities. Therefore, it is necessary to use other methods of pain assessment (e.g., pupillometry) in this group. Optimizing analgesia (including through nurse-controlled analgesia) and sedation in ICU patients, especially during painful procedures, is essential.

## 8. Conclusions

The results of the study indicate that some nursing procedures commonly used in the ICU are a source of pain, also in patients undergoing deep sedation and receiving analgesics. The BPS and CCPOT scales are useful tools for assessing the occurrence of pain in this group of patients.

## Figures and Tables

**Figure 1 ijerph-19-10894-f001:**
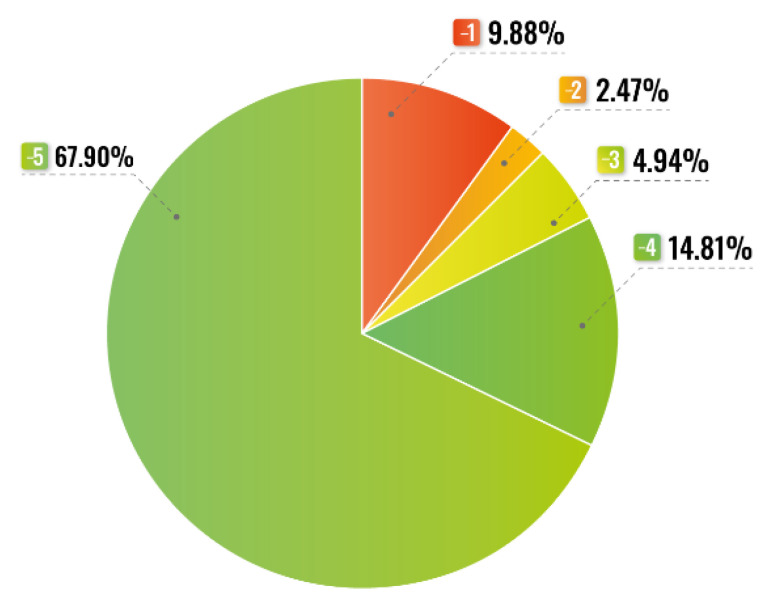
RASS scores in the study group (*n* = 81).

**Figure 2 ijerph-19-10894-f002:**
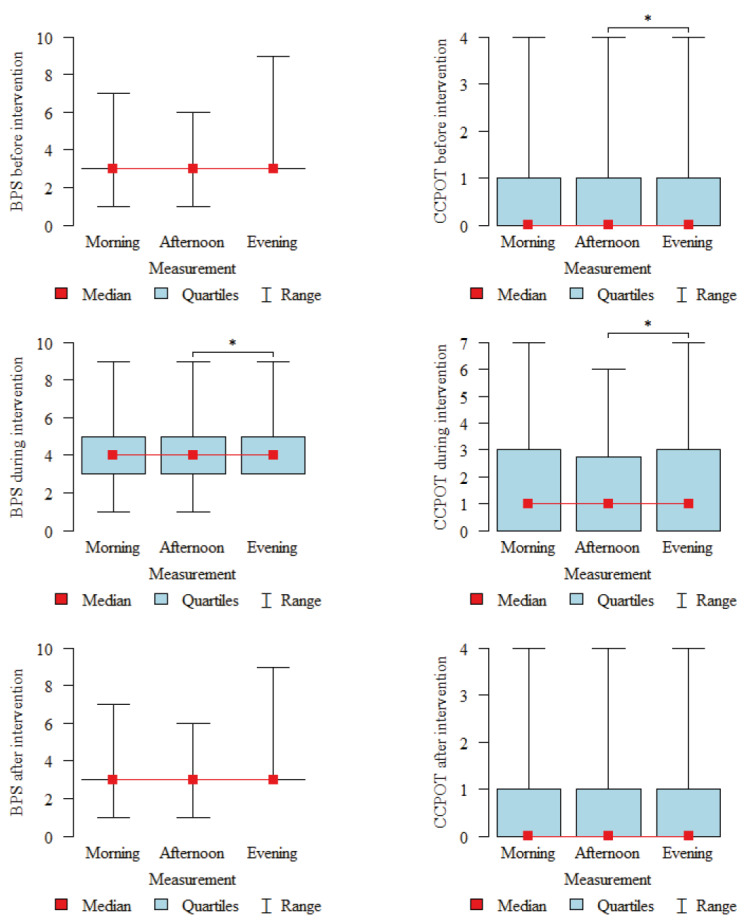
Mean BPS and CCPOT scores at different times of day (data from1005 measurements). * *p* < 0.05; *p*–Friedman’s test + post-hoc analysis (Wilcoxon paired *t* tests with Bonferroni correction). “Interventions” refer to painful nursing procedures (endotracheal suctioning, patient turning, dressing changes).

**Table 1 ijerph-19-10894-t001:** Average doses of analgesics and sedatives used in the study group.

Type of Drug	Dose i.v.
Oxycodone	0.5–9.5 mg/h
Propofol	10–300 mg/h
Midazolam	1–30 mg/h
Dexmedetomidine	0.008–0.18 mg/h
Thiopental	70–240 mg/h
Ketamine	10–250 mg/h

**Table 2 ijerph-19-10894-t002:** BPS and CCPOT score ranges in the study group (data from 1005 measurements).

Pain Scale Scores	Before Intervention *n* (%)	During Intervention *n* (%)	After Intervention *n* (%)
**BPS**			
<5 points	959 (95.42%)	678 (67.46%)	937 (93.23%)
≥5 points	46 (4.58%)	327 (32.54%)	68 (6.77%)
**CCPOT**			
<3 points	970 (96.52%)	706 (70.25%)	958 (95.33%)
≥3 points	35 (3.48%)	299 (29.75%)	47 (4.67%)

**Table 3 ijerph-19-10894-t003:** Mean BPS and CCPOT scores in all measures (*n* = 1005).

	Before Intervention (BI)	During Intervention (I)	After Intervention (AI)	*p*
**BPS**				
mean ± SD	3.2 ± 0.58	4.12 ± 1.37	3.25 ± 0.66	*p* < 0.001
median	3	4	3	
quartiles	3–3	3–5	3–3	I > BI,AI
**CCPOT**				
mean ± SD	0.47 ± 0.78	1.66 ± 1.73	0.52 ± 0.85	*p* < 0.001
median	0	1	0	
quartiles	0–1	0–3	0–1	I > BI,AI

*p*—Friedman’s test + post-hoc analysis (Wilcoxon paired t tests with Bonferroni correction).

**Table 4 ijerph-19-10894-t004:** Correlations between RASS scores and BPS, CCPOT scores.

Parameter	RASS
	Spearman Correlation Coefficient (R)
BPS before intervention	R = 0.279, *p* = 0.012 *
BPS during intervention	R = 0.444, *p* < 0.001 *
BPS after intervention	R = 0.293, *p* = 0.008 *
CCPOT before intervention	R = 0.438, *p* < 0.001 *
CCPOT during intervention	R = 0.556, *p* < 0.001 *
CCPOT after intervention	R = 0.446, *p* < 0.001 *

R—value of the Spearman’s rank correlation test; * statistically significant relationship (*p* < 0.05).

**Table 5 ijerph-19-10894-t005:** Correlations between BPS and CCPOT scores.

Measurement	Spearman Correlation Coefficient (R) between BPS and CCPOT	*p*
Before intervention	0.695	*p* < 0.001 *
During intervention	0.907	*p* < 0.001 *
After intervention	0.622	*p* < 0.001 *

R—value of the Spearman’s rank correlation test; *p*—test probability index. * statistically significant relationship (*p* < 0.05).

**Table 6 ijerph-19-10894-t006:** Correlations between patients’ age and BPS, CCPOT scores (data from 1005 measurements).

Scales Scores	Correlations with Age	
	(R)	*p*
BPS before interventions	−0.073	*p* = 0.027 *
BPS during interventions	0.04	*p* = 0.222
BPS after interventions	−0.029	*p* = 0.379
CCPOT before interventions	−0.011	*p* = 0.731
CCPOT during interventions	0.061	*p* = 0.062
CCPOT after interventions	−0.007	*p* = 0.839

R—value of the Spearman’s rank correlation test; *p*—test probability index. * statistically significant relationship (*p* < 0.05).

**Table 7 ijerph-19-10894-t007:** BPS and CCPOT scores versus length of hospitalization (data from 1005 measures).

Scales Scores	Correlations with Length of Hospitalization	
	R	*p*
BPS before interventions	0.055	*p* = 0.09
BPS during interventions	0.092	*p* = 0.005 *
BPS after interventions	0.11	*p* = 0.001 *
CCPOT before interventions	0.054	*p* = 0.095
CCPOT during interventions	0.093	*p* = 0.004 *
CCPOT after interventions	0.083	*p* = 0.01 *

R—value of the Spearman’s rank correlation test; *p*—test probability index. * statistically significant relationship (*p* < 0.05).

## Data Availability

The data presented in this study are available on request from the corresponding author.

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
