# Peer review of "Pain Assessment with the BPS and CCPOT Behavioral Pain Scales in Mechanically Ventilated Patients Requiring Analgesia and Sedation"

_ijerph, 2022, doi:10.3390/ijerph191710894_

Round 1

Reviewer 1 Report

Complete the description of the analgesia scheme.

Correctly define the method of selecting the study group.

Other comments are included in the attached file.

Author Response

Dear Rewiever,

Thank you very much for all comments which will allow us to improve our manuscript.  We greatly value the time and effort you put into reviewing our manuscript. 

The study analyzed the usefulness of the BPS Behavioral Pain Scale (BPS) and Critical Care Pain Observation Tool (CCPOT) scales in assessing pain among patients with varying degrees of sedation. The topic of the study is very interesting and important from a practical point of view. However. Among limitations, the reported study has been conducted in one center, focused on a single sedation regimen, omitting analgesic monitoring.

Thank you for your positive feedback. As regards analgesic monitoring, we have extended the explanation of sedation and analgesic protocol including monitoring in the main text.

Generally:

No single selected analgosedation protocol was used in the study group. Analgesia and sedation in patients were considered individually and adjusted to the patient's clinical condition. The Trauma Centre for Emergency and Disaster Medicine, where we conducted our study, was the only 'trauma centre' in Poland, which treated the most severely ill patients - most often with multiple and multi-organ trauma, requiring highly specialised monitoring, treatment (usually surgical interventions) and nursing care (third degree of reference).

The multitrauma and very severe clinical conditions of the patients influenced the depth of sedation used, which is not contrary to the guidelines  (i.e.PAD 2013 recommendation) [Barr J.et al. 2013] that were in force at the time of the study.

Deep sedation (according to eCASH guidelines) [Vincent et al.] was used in the following patient groups: with severe respiratory failure and dyssynchrony, in surgical treatment requiring absolute immobilisation and in cases of severe brain injury with intracranial hypertension.

At the same time, non-pharmacological preventive strategies [Baron et al.] and analgesic treatment using algorithms developed by Polish pain relief teams were applied [Misiołek].

Approximately 80% of patients in Poland admitted to the ICU are mechanically ventilated. The trend in Poland is towards minimising sedation with the duration of patients' ICU stay rather than introducing breaks in sedation [Kotfis et al.].

The research team did not intervene in the treatment regimen of the study patients. Prior to the presented study, monitoring of sedation and analgesia with standardised tools had not been routinely used. We did so only during our study, and only recently a systematic pain assessment using CCPOT has been introduced.

  • Barr, J.; Fraser, G.L.; Puntillo, K., et al. Clinical practice guidelines for the management of pain, agitation and delirium in adult patients in the Intensive Care Unit. CCM Journal 2013, 41, 263-306.
  • Vincent, J.L. Optimizing sedation in the ICU: the eCASH concept. Signa Vitae 2017, 13, 10-13.
  • Baron R., Binder A., Biniek R., Braune S., et al. On behalf of DAS-Taskforce 2015: Evidence and consensus based guideline for the management of delirium, analgesia, and sedation in intensive care medicine. Revision 2015 (DAS-Guideline 2015) – short version. GMS German Medical Science 2015; 13: 1-42.
  • Misiołek H., Zajączkowska R., Daszkiewicz A., Woroń J., Dobrogowski J., Wordliczek J. , Owczuk R.: Postępowanie w bólu pooperacyjnym 2018 — stanowisko Sekcji Znieczulenia Regionalnego i Terapii Bólu Polskiego Towarzystwa Anestezjologii i Intensywnej Terapii, Polskiego Towarzystwa Znieczulenia Regionalnego i Leczenia Bólu, Polskiego Towarzystwa Badania Bólu oraz Konsultanta Krajowego w dziedzinie anestezjologii i intensywnej terapii. Anestezjologia Intensywna Terapia 2018: 50 (3); 175–203.
  • Kotfis, K.; Zegan-Barańska, M.; Żukowski, M.; Kusza, K.; Kaczmarczyk, M.; Ely, EW. Multicenter assessment of sedation and delirium practices in the intensive care units in Poland - is this common practice in Eastern Europe? BMC Anesthesiol. 2017, 2, 120.

Detailed comments

Introduction

Gives a good introduction to the issue under study, although it focuses excessively on the effects of sedation.

Indeed, in the introduction we devoted much attention to sedation and its effects, due to the presence of the so-called ICU triad which can be prevented by effective analgosedation, and because the group we studied included mainly deeply sedated patients. We wanted to point out that in some groups of patients such sedation is necessary despite potential side effects and pain assessment is much more difficult. In our study, we were interested in whether this group of patients also expressed behaviours indicating pain (examined with the BPS and CCPOT scales). However, following your comment, we have added a paragraph in the main text on the consequences of untreated pain and the importance of routine pain monitoring in the ICU what is important in our study. We also resigned from the detailed description of negative effects of deep sedation as you suggested because it was not directly related to the purpose of our study. However, we left part of the description as it was pointed out as an advantage of the work by another reviewer.

Methods

However, the observational study, which has the advantage of being assessed by three independent observers, did not include a detailed description of the analgesia regimen used and, most importantly, the time elapsed between analgesic administration and the interventions in which pain was assessed.

Thank you for this important comment. Analgesics were individually tailored to patients needs, administered by continuous infusion. For the purposes of the study, no additional boluses of analgesics were administered during observation. This was added as an explanation in the „Material and methods” section in the description of analgesia and sedation regimen. The answer for this comment can be also found in the reply to comment 1.

Results

The results of the study are reproducible based on the details provided in the research methods section, presented in tables and figures are easy to interpret and understand. Data are adequately and consistently interpreted throughout the manuscript. Whenever the existing correlation between the level of sedation and the pain score was indicated, it was missing an analysis of the correlation between the dose and time of analgesia administration and the pain intervention performed.

As explained in the previous comment, the analgesics were administered by continuous infusions, with no additional boluses during observation period. Therefore we have not analysed correlations between time of analgesia administration and pain scores during painful procedures. Moreover, large differentiation among patients in terms of disease entities, the related individual selection of analgesia and the interactions of sedatives and analgesics made it impossible to analyze the relationships between the type and doses of analgesics and the results of the BPS, CCPOT and RASS scales. We mentioned it in the section “Study limitations”.

Discussion

The discussion is overly exclusively, focusing on reports comparing different approaches to sedation of critically ill patients. Little attention was paid to the importance of monitoring analgesia in the care of critically ill mechanically ventilated patients.

Analgesia and sedation are interrelated and in our patients, they have always been used together. Sedation tends to be used in the ICUs much more than the recommendations say and depth of sedation can affect pain assessment, therefore we described this in the discussion. However,we agree with your comment that more attention should have been paid to the importance of monitoring analgesia. We wrote about it in the section "Implications to practice". In the revised version, we have also included this issue in discussion and shortened a bit issues related to approaches to sedation.

Conclusions consistent with the evidence presented are adequate for the stated goals of the study. In conclusion - the manuscript is clear and presented in a well-organized manner, focused on an issue of great importance to clinical practice.

Thank you very much for this comment.

Reviewer 2 Report

The paper address an important issue and it is written clearly, covers important advantage and pitfalls of sedation and analgesia in ICU patients.

However, I fear that it does not add to much to the scientific available evidence. It is quite simple to understand that pain is well controlled in deep sedated patients at rest, and that it increases during painful procedures. However, what is the clinical significance of 1 point change?

Moreover, the authors stated that they consider painful procedures tracheal suctioning or dressing changes, but I guess that this 2 procedures have not the same pain burden in critically ills, as other authors report. So maybe they mix different procedures but not all of them are comparable. 

In any case, the authors provided data about analgesia , that I guess is performed in a continuous way, but what about these procedures? Did they use some more analgesics (like bolus)? Did they increase sedation?

As they stated , in many cases the pain scores returned to baseline, and I guess in few minutes (to date, no data were provided) Once again, had these results  a clinical significance? Maybe the authors should discuss these points.

Finally, the cumulative effect point at the end of the  discussion needs further insights.

The implication to practice is interesting but completely different from the results obtained, and it confirms what we already know. So, once again, I'm not sure the present paper adds something new to the available evidence.

Author Response

Dear Rewiever,

Thank you very much for all comments which will allow us to improve our manuscript.  We greatly value the time and effort you put into reviewing our manuscript. 

The paper address an important issue and it is written clearly, covers important advantage and pitfalls of sedation and analgesia in ICU patients.

However, I fear that it does not add to much to the scientific available evidence. It is quite simple to understand that pain is well controlled in deep sedated patients at rest, and that it increases during painful procedures. However, what is the clinical significance of 1 point change?

The purpose of our study was not to prove that deeply sedated patients have well-controlled pain at rest and that it increases during painful procedures, but to show that the BPS and CCPOT scales are useful in assessing the occurrence of pain in deeply sedated patients (patients manifest pain non-verbally according to criteria of BPS and CCPOT). Previous studies have mainly analyzed the usefulness of the scales in non-sedated or mildly sedated patients. Reports on the assessment of pain in deeply sedated patients are limited and their results contradictory. A new aspect of our study is that it shows that the BPS and CCPOT scales are also useful in assessing pain in this group of patients.

Although BPS and CCPOT scores increased by an average of 1 point (which was statistically significant) - this showed that also deeply sedated patients manifested pain during painful procedures and this was visible to the investigators.

Moreover, the authors stated that they consider painful procedures tracheal suctioning or dressing changes, but I guess that this 2 procedures have not the same pain burden in critically ills, as other authors report. So maybe they mix different procedures but not all of them are comparable. 

According to Puntillo et al*., who conducted prospective, cross-sectional, multicenter, multinational study of pain intensity associated with 12 procedures the two  procedures we have analyzed (endotracheal suctioning and turning) were comparable with respect to pain intensity during the procedure. Wound care was a little bit less painful in their study but in our patients, the wounds were usually surgical wounds in the early post-operative period so we assumed that the pain related to dressing change was similar to the other two procedures.

Also Rijkenberg  & van der Voort [**] agree with this approach.

Apart from that in the „CPOT Directives of Use” (Adapted from Gélinas et al., AJCC 2006; 15(4):420-427***) the authors write: „1. The patient must be observed at rest for one minute to obtain a baseline value of the CPOT. 2. Then, the patient should be observed during nociceptive procedures known to be painful (e.g. turning, wound care) to detect any changes in the patient’s behaviors to pain”

*Puntillo, K.A.;  Max, A.; Timsit, J.F.; et al. Determinants of procedural pain intensity in the intensive care unit. The Europain study. Am. J. Respir. Crit. Care Med. 2014, 189, 39-47.

** Rijkenberg S, van der Voort PH. Can the critical-care pain observation tool (CPOT) be used to assess pain in delirious ICU patients? J Thorac Dis. 2016 May;8(5):E285-7. doi: 10.21037/jtd.2016.03.32. PMID: 27162683; PMCID: PMC4842811.

***CPOT Directives of Use (Adapted from Gélinas et al., AJCC 2006; 15(4):420-427 https://blog.summit-education.com/wp-content/uploads/CPOT.pdf

In any case, the authors provided data about analgesia, that I guess is performed in a continuous way, but what about these procedures? Did they use some more analgesics (like bolus)? Did they increase sedation?

Thank you for this important comment. We have supplemented the information in the text that the analgesics were administered in continuous infusions, in doses individually tailored to the patients' needs, and no additional boluses were administered during observations. Outside observation, however, modifications of doses or boluses were used if the patient required them during care.

As they stated , in many cases the pain scores returned to baseline, and I guess in few minutes (to date, no data were provided. Once again, had these results  a clinical significance? Maybe the authors should discuss these points.

We did not analyze how long it took for the pain to return to baseline, because it was not the aim of our study. The purpose of the study was not also to examine how painful the procedures are, but whether patients manifest any signs of pain during these painful procedures, even if they are deeply sedated and whether they differ from behaviour at rest. Thus, we wanted to find out whether BPS and CCPOT can be used in deeply sedated patients. As for the time during which the observations were conducted, we have provided information in the text that „rest meant no medical or nursing intervention for at least 30 minutes (…) The evaluation after the intervention was carried out for dozen to several dozen minutes without stimulation from external stimuli”. In the revised version we added explanation ones again before tab.3.

Finally, the cumulative effect point at the end of the  discussion needs further insights.

By cumulative effect we mean sum of negative stimuli (e.g. fatigue, sleep deprivation, excess stimuli from the ICU environment) during the daytime hours and along with the length of stay in the ward which may affect pain. We added this explanation in the end of the discussion.

The implication to practice is interesting but completely different from the results obtained, and it confirms what we already know. So, once again, I'm not sure the present paper adds something new to the available evidence.

We answer for this comment is in point 1.

Reviewer 3 Report

1.       Please clearly state in the introduction the main and secondary outcomes followed and follow them in the results and discussion section

2.       Please specify in the main text the sedation regimens. It is confusing and discordant with the informations offered in the tables. Do you have any sedation protocol? Please specify

3.       Please reconsider the design of the fig.1 as parts of a whole (i.e., pie)

4.       MAJOR: Please add/specify the characteristics of the control group. Do you have a control group? It is unclear.

5.       The randomization of the monitoring time at the experts discretion is, in my opinion, inappropriate

6.       The classification of the pain into low/moderate/acute is inadequate. Please reconsider

7.       MAJOR: Please divide your patients into surgical and non-surgical and perform a subgroup analysis. Taken all together, is, however, not an option for this kind of study.

8.       Could you provide the statistical power and the normality tests used?

9.       Please clearly define the study groups

1.   Rows 117-118. Please specify more accurate which parameters for which values

1.   The results section should offer informations about the main and secondary outcomes, as defined in introduction. Please reconsider.

1.   Please provide the manufacturer of the medication used

1.   Row 146- you mentioned again the study groups. Are there other groups as well?

1.   The table 3 provides no additional information with regard to table 2. Please reconsider

1.   Other outcomes not included in the introduction section: rows 182-185, 185-195

1.   Rows 206-208- Fig.2- please add p values, please specify which kind of interventions you referred to

1.   Dexdor and thiopental appears only in tables. Please specify the exact sedation protocol you used

Author Response

Dear Rewiever,

Thank you very much for all comments which will allow us to improve our manuscript.  We greatly value the time and effort you put into reviewing our manuscript. 

Please clearly state in the introduction the main and secondary outcomes followed and follow them in the results and discussion section.

We would be happy to address this comment, but we are not sure what it refers to. We would be gratefu if you could specify what you mean when you write about „the main and secondary outcomes” in the introduction.

Please specify in the main text the sedation regimens. It is confusing and discordant with the informations offered in the tables. Do you have any sedation protocol? Please specify.

Thank you for this comment. We have extended the explanation of sedation and analgesic protocol in the main text.

Generally:

No single selected analgosedation protocol was used in the study group. Analgesia and sedation in patients were considered individually and adjusted to the patient's clinical condition. The Trauma Centre for Emergency and Disaster Medicine, where we conducted our study, was the only 'trauma centre' in Poland, which treated the most severely ill patients - most often with multiple and multi-organ trauma, requiring highly specialised monitoring, treatment (usually surgical interventions) and nursing care (third degree of reference).

The multitrauma and very severe clinical conditions of the patients influenced the depth of sedation used, which is not contrary to the guidelines  (i.e.PAD 2013 recommendation) [Barr J.et al. 2013] that were in force at the time of the study.

Deep sedation (according to eCASH guidelines {Vincent et al.] was used in the following patient groups: with severe respiratory failure and dyssynchrony, in surgical treatment requiring absolute immobilisation and in cases of severe brain injury with intracranial hypertension.

At the same time, non-pharmacological preventive strategies [Baron et al.] and analgesic treatment using algorithms developed by Polish pain relief teams were applied [Misiołek].

Approximately 80% of patients in Poland admitted to the ICU are mechanically ventilated. The trend in Poland is towards minimising sedation with the duration of patients' ICU stay rather than introducing breaks in sedation [ Kotfis, no. 15].

The research team did not intervene in the treatment regimen of the study patients or monitor the effectiveness of the pain management (continuous observation of the patients), but only investigated the occurrence of pain at different times of hospitalisation of the eligible patients.

  • Barr, J.; Fraser, G.L.; Puntillo, K., et al. Clinical practice guidelines for the management of pain, agitation and delirium in adult patients in the Intensive Care Unit. CCM Journal 2013, 41, 263-306.
  • Vincent, J.L. Optimizing sedation in the ICU: the eCASH concept. Signa Vitae 2017, 13, 10-13.
  • Baron R., Binder A., Biniek R., Braune S., et al. On behalf of DAS-Taskforce 2015: Evidence and consensus based guideline for the management of delirium, analgesia, and sedation in intensive care medicine. Revision 2015 (DAS-Guideline 2015) – short version. GMS German Medical Science 2015; 13: 1-42.
  • Misiołek H., Zajączkowska R., Daszkiewicz A., Woroń J., Dobrogowski J., Wordliczek J. , Owczuk R.: Postępowanie w bólu pooperacyjnym 2018 — stanowisko Sekcji Znieczulenia Regionalnego i Terapii Bólu Polskiego Towarzystwa Anestezjologii i Intensywnej Terapii, Polskiego Towarzystwa Znieczulenia Regionalnego i Leczenia Bólu, Polskiego Towarzystwa Badania Bólu oraz Konsultanta Krajowego w dziedzinie anestezjologii i intensywnej terapii. Anestezjologia Intensywna Terapia 2018: 50 (3); 175–203.
  • Kotfis, K.; Zegan-Barańska, M.; Żukowski, M.; Kusza, K.; Kaczmarczyk, M.; Ely, EW. Multicenter assessment of sedation and delirium practices in the intensive care units in Poland - is this common practice in Eastern Europe? BMC Anesthesiol. 2017, 2, 120.

Please reconsider the design of the fig.1 as parts of a whole (i.e., pie)

We have changed the graph to a pie chart as you suggested.

MAJOR: Please add/specify the characteristics of the control group. Do you have a control group? It is unclear.

We do not have a control group in our study. We designed the study so that in a group of patients who met certain criteria (described in the section “Material and methods”), we assessed patients behaviours expressing pain using the BPS and CCPOT scales three times a day, during rest, nursing interventions and after an intervention. We compared whether the scores of the scales differed significantly during the intervention compared to the baseline values and the time after the intervention (pre-during-post tests), whether there was a strong correlation between the scores of the scales, what was the relationship between the degree of sedation of the patients and the scores of BPS and CCPOT, etc. We think that for the purpose of the study, a control group was not necessary.

The randomization of the monitoring time at the experts discretion is, in my opinion, inappropriate

The hours of observation of patients were more or less fixed and consciously adjusted by us to the specific work of the team (nursing activities, diagnostic procedures, etc.). In the morning, afternoon and evening at the times we specified in the „Material and methods”, it was possible to examine patients both at rest, then during nursing activities, and at specific times after interventions.

The classification of the pain into low/moderate/acute is inadequate. Please reconsider

We previously considered (according to Severgnini et al.) a BPS score 3-4 and CCPOT score 0-2 like absence of pain; a BPS score 5-7 and CCPOT score 3-4 like moderate pain; and a BPS score 8-12 and CCPOT score 5-8 like severe pain.

However, we agree with your comment and in agreement with the literature [Chanques at al; Gelinas et al., Kotfis et al.], the scores BPS≥5 and  CCPOT ≥3 were used as a cutoff value to determine patients with pain/no pain with no classification of severity of pain. We changed therefore tab.2 and some fragments in the main text.

  • Severgnini, P.; Pelosi, P.; Contino, E.; Serafinelli, E.; Novario, R.; Chiaranda, M. Accuracy of critical care pain observation tool and behavioral pain scale to assess pain in critically ill conscious and unconscious patients: prospective, observational study. J. Intensive Care. 2016, 4, 68–68.
  • Chanques G., Tarri T., Ride A., Prades A., , De Jong A., Carr J., Molinari N., Jaber S. Analgesia nociception index for the assessment of pain in critically ill patients: a diagnostic accuracy study. British Journal of Anaesthesia, 2017; 119 (4): 812–20.
  • Gélinas C, Fillion L, Puntillo KA, et al. Validation of the critical-care pain observation tool in adult patients. Am J Crit Care. 2006; 15(4): 420–427, indexed in Pubmed: 16823021. 
  • Kotfis, K.; Zegan-Barańska, M.; Strzelbicka, M.; Safranow, K.; Żukowski, M.; Ely, W. the POL-CPOT Study Group. Validation of the Polish version of the Critical Care Pain Observation Tool (CPOT) to assess pain intensity in adult, intubated intensive care unit patients: the POL-CPOT study. Arch. Med. Sci. 2018, 14, 880–889.

MAJOR: Please divide your patients into surgical and non-surgical and perform a subgroup analysis. Taken all together, is, however, not an option for this kind of study.

We deliberately did not divide patients in subgroups of surgical and non-surgical patients because we were interested in the occurrence of pain during procedures regardless of whether the patient was surgical or not. The ICU in which we conducted the study was highly specialised (the Trauma Centre for Emergency and Disaster Medicine), which treated the most seriously ill patients - usually those with multiple and multi-organ injuries who experienced similar pain regardless of whether they were treated surgically or conservatively. Non-verbal pain expression, which is the criterion for assessing pain in the BPS and CCPOT scales, was considered by us to be the most important criterion. Sometimes patients were first treated conservatively and then surgically so it would be difficult to classify them into one of the proposed subgroups (surgical – non-surgical).

Could you provide the statistical power and the normality tests used?

The normality of the distribution was not checked, as it is evident that  all quantitative variables (raw scores of BPS, CCPOT and RASS) were not normally distributed, by definitione - raw scores of BPS, CCPOT and RASS could take only few distinct values while a normal distribution can only have variables that take a lot of different values.

Please clearly define the study groups

There was one study group - adult ICU patients unable to self-report pain, requiring sedation (RASS less than or equal to -1) and analgesia, mechanically ventilated, hemodynamically stable, and staying in the ICU for at least 48 hours, excluding patients with medical conditions that made them unable to express pain non-verbally. A description of the study group, as well as the inclusion and exclusion criteria, are described in the section "Material and Methods."

Rows 117-118. Please specify more accurate which parameters for which values

Below there is a summary of which statistical tests were used in the analysis. We have included explanations in the main text below each table and figure.

Number of table/figure

Parametric test

Non-parametric test

Table 2. BPS and CCPOT score ranges in the study group

x

 Descriptive statistics

Tabele 3. Mean BPS and CCPOT scores in all measures

x

p - Friedman’s test +post-hoc analysis (Wilcoxon paired t tests with Bonferroni correction).

Table 4. Correlations between RASS scores and BPS, CCPOT scores.

x

Spearman correlation coefficient

Table 5. Correlations beetwen BPS and CCPOT scores.

x

R – value of the Spearman's rank correlation test;

Table 6. Correlations between patients' age and BPS, CCPOT scores

x

R – value of the Spearman's rank correlation test

Table 7. BPS and CCPOT scores versus length of hospitalization

x

R – value of the Spearman's rank correlation test

Figure 1. RASS scores in the study group (n=81).

x

  Descriptive statistics

Figure 2. Mean BPS and CCPOT scores at different times of day

x

p – Friedman’s test + post-hoc analysis (Wilcoxon paired t tests with Bonferroni correction)

The results section should offer informations about the main and secondary outcomes, as defined in introduction. Please reconsider.

We would be happy to address this comment, but we are not sure what it refers to. We would be gratefu if you could clarify it.

Please provide the manufacturer of the medication used

We decided to standardise the names of medicines in the manuscript (previously this was not consistent) by giving the active substance, which is identifiable to readers in different countries. Drugs from different manufacturers were used on the ward at different times.

Row 146- you mentioned again the study groups. Are there other groups as well?

In the row 146, as well as 148 and table 1.we write about „study group” not „study groups”

The table 3 provides no additional information with regard to table 2. Please reconsider

After reading the comment of Reviewer 2 we have changed a little bit table 3. It shows now the percetage of patients with and without pain (assessed by two scales), whereas data in the table 2 shows the statistically  significant differences between BPS and CCPOT scores during painful interventions in comparison with the time before/after them. We would like to leave both tables if possible.

Other outcomes not included in the introduction section: rows 182-185, 185-195

We would be happy to address this comment, but we are not sure what it refers to. We would be gratefu if you could clarify it.

Rows 206-208- Fig.2- please add p values, please specify which kind of interventions you referred to

The p-values were plotted on the graph. Interventions refer to the painful interventions described in the manuscript (tracheal suctioning, patient turning and wound dressing changes). Explanation was added under the fig.2

Dexdor and thiopental appears only in tables. Please specify the exact sedation protocol you used

Thank you for this comment. We have extended the explanation of sedation and analgesic protocol in the main text.

Generally:

No single selected analgosedation protocol was used in the study group. Analgesia and sedation in patients were considered individually and adjusted to the patient's clinical condition. The Trauma Centre for Emergency and Disaster Medicine, where we conducted our study, was the only 'trauma centre' in Poland, which treated the most severely ill patients - most often with multiple and multi-organ trauma, requiring highly specialised monitoring, treatment (usually surgical interventions) and nursing care (third degree of reference).

The multitrauma and very severe clinical conditions of the patients influenced the depth of sedation used, which is not contrary to the guidelines  (i.e.PAD 2013 recommendation) [Barr J.et al. 2013] that were in force at the time of the study.

Deep sedation (according to eCASH guidelines) [Vincent et al.] was used in the following patient groups: with severe respiratory failure and dyssynchrony, in surgical treatment requiring absolute immobilisation and in cases of severe brain injury with intracranial hypertension.

At the same time, non-pharmacological preventive strategies [Baron et al.] and analgesic treatment using algorithms developed by Polish pain relief teams were applied [Misiołek].

Thiopental was used in only 4 patients with craniocerebral trauma (acute subdural haematoma, supratentorial haematoma, cerebral contusion, post-traumatic tSAH; multifracture fracture of the iliac plate and shaft, pubic bone; lung injury with bilateral multiple rib fracture, lung and heart contusion and respiratory failure; fracture of the sternum and right scapula) in which it was necessary to keep the patient in a pharmacological coma.

Dexmedetomidine was used in postoperative pharmacotherapy by continuous i.v. infusion in combination with a strong opioid, i.e. oxycodone, at a dose determined by a titration procedure according to Polish guidelines [Kotfis et al.] for pain relief after surgery with significant and extensive tissue trauma.

We have decided to omit this last description (Thioental, dexmedetomidyne) in the text as it is too detailed.

  • Barr, J.; Fraser, G.L.; Puntillo, K., et al. Clinical practice guidelines for the management of pain, agitation and delirium in adult patients in the Intensive Care Unit. CCM Journal 2013, 41, 263-306.
  • Vincent, J.L. Optimizing sedation in the ICU: the eCASH concept. Signa Vitae 2017, 13, 10-13.
  • Baron R., Binder A., Biniek R., Braune S., et al. On behalf of DAS-Taskforce 2015: Evidence and consensus based guideline for the management of delirium, analgesia, and sedation in intensive care medicine. Revision 2015 (DAS-Guideline 2015) – short version. GMS German Medical Science 2015; 13: 1-42.
  • Misiołek H., Zajączkowska R., Daszkiewicz A., Woroń J., Dobrogowski J., Wordliczek J. , Owczuk R.: Postępowanie w bólu pooperacyjnym 2018 — stanowisko Sekcji Znieczulenia Regionalnego i Terapii Bólu Polskiego Towarzystwa Anestezjologii i Intensywnej Terapii, Polskiego Towarzystwa Znieczulenia Regionalnego i Leczenia Bólu, Polskiego Towarzystwa Badania Bólu oraz Konsultanta Krajowego w dziedzinie anestezjologii i intensywnej terapii. Anestezjologia Intensywna Terapia 2018: 50 (3); 175–203.
  • Kotfis, K.; Zegan-Barańska, M.; Żukowski, M.; Kusza, K.; Kaczmarczyk, M.; Ely, EW. Multicenter assessment of sedation and delirium practices in the intensive care units in Poland - is this common practice in Eastern Europe? BMC Anesthesiol. 2017, 2, 120.

Round 2

Reviewer 3 Report

thank you for answering all the mentioned problems